# *TemporalBench*: Benchmarking Fine-grained Temporal Understanding for Multimodal Video Models

## Abstract

Understanding fine-grained temporal dynamics is crucial for video understanding. Yet, popular video benchmarks, such as MSRVTT and TGIF, often fail to effectively evaluate AI models' temporal reasoning abilities due to the lack of fine-grained temporal annotations. As a result, text-based models, leveraging strong language priors, often perform comparably to video models, and image-trained models have been reported to outperform their video-trained counterparts on MSRVTT and TGIF. This paper introduces a new *TemporalBench* benchmark for fine-grained temporal event understanding in videos. TemporalBench, sourced from a diverse video datasets, consists of ∼10K pairs of video description questions, derived from ∼2K high-quality human-annotated video captions. Uniquely, our benchmark provides fine-grained temporal annotations to evaluate models' temporal reasoning abilities. Our results show that state-of-the-art models like GPT-4o achieve only 38.0% multiple binary QA accuracy on TemporalBench, demonstrating a significant human-AI gap in temporal understanding. We hope that TemporalBench is instrumental to fostering research on improving models' temporal reasoning capabilities.

## 1 Introduction

The ability to understand and reason about events in videos is a crucial aspect of artificial intelligence, with applications ranging from activity recognition and long-term action anticipation to perception for autonomous driving and robotics. Recently, there has been an emergence of highly capable multimodal generative models, including proprietary ones such as GPT-4o (OpenAI, 2024) and Gemini (Gemini Team, 2024) as well as open-sources ones (Liu et al., 2023a; Zhu et al., 2024b; Bai et al., 2023), that have demonstrated impressive results on existing video benchmarks (Xu et al., 2016; Chen & Dolan, 2011; Yu et al., 2019a; Mangalam et al., 2024). However, these benchmarks often do not truly evaluate the abilities of the aforementioned models to understand video content due to their generally *coarse-grained* annotations.

The lack of fine-grained temporal details in the annotations often leads to existing video understanding benchmarks suffering from a strong language prior bias. This is similar to observations in visual question answering with images (Antol et al., 2015). For example, prior works (Tan et al., 2024; Li et al., 2023a) show that language models such as Flan-T5 (Chung et al., 2024) and Llama-2/3 (Touvron et al., 2023) perform comparably to video models on EgoSchema (Mangalam et al., 2024) and Seed-Bench (Li et al., 2023a) without using any information from videos. Furthermore, the lack of fine-grained temporal details often results in the single frame bias of current video understanding benchmarks (Lei et al., 2023). These benchmarks are often biased toward spatial reasoning, where static information from a single frame suffices to achieve high performance. They often fail to test a model's ability to reason about temporal sequences, leading to inflated evaluations of AI models that are not genuinely capable of understanding temporal events. Specifically, vision-language models (VLMs) (Liu et al., 2024a;b) that are trained on image-level datasets, including FreeVA (Wu, 2024), IG-VLM (Kim et al., 2024) and $M^3$ (Cai et al., 2024b), often outperform their video counterparts on popular video question answering benchmarks such as MSRVTT (Xu et al., 2016), MSVD (Xu et al., 2017), and TGIF (Jang et al., 2017).

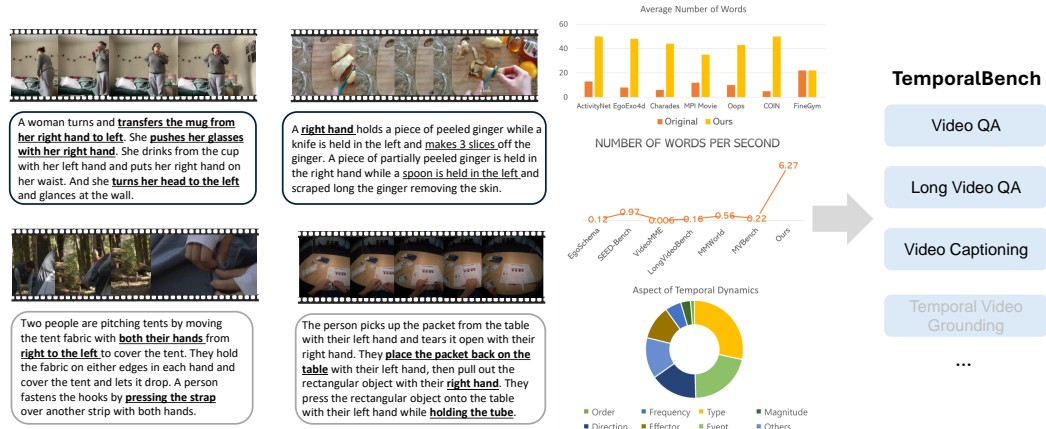

Figure 1: **The tasks of *TemporalBench***. *TemporalBench* starts from fine-grained video descriptions and supports diverse video understanding tasks including video QA, video captioning, long video understanding, *etc*. It differs from existing benchmarks by the average number of words per video (middle top), word density (center) and the coverage of various temporal aspects (middle bottom).

To address this limitation, we propose *TemporalBench* (Figure 1), a new video understanding benchmark that evaluates multimodal video models on understanding fine-grained activities, and consists of ~**10k** question and answer pairs curated from ~**2k** high-quality human-annotated captions with rich activity details. Unlike static image-based tasks, video understanding requires models to reason effectively about both spatial and temporal information. The temporal dynamics inherent in videos introduce significant complexity, as actions and events often unfold over time and cannot be captured in a single frame. With this in mind, we designed our benchmark to focus on areas where current models often struggle, emphasizing annotations related to long-range dependencies, fine-grained visual observations, and event progression.

As shown in Figure 2, we first collect video clips from existing video grounding benchmarks that span diverse domains, including procedural videos (Tang et al., 2019), human activities (Krishna et al., 2017; Gao et al., 2017) and ego-centric videos (Grauman et al., 2024). The positive captions include *rich* and *fine-grained* details about actions and activities, which are annotated by highly qualified Amazon Mechanical Turk (AMT) workers and authors of this paper. Then, we generate the negative captions with respect to the actions using powerful Large Language Models (LLMs) and filter them according to our defined rules. Our resulting *TemporalBench* contains 10K video descriptions and matching questions of high quality. Furthermore, the rich temporal context of annotations in our diverse corpus creates a solid foundation for the development of additional benchmarks in related tasks such as spatio-temporal localization and causal inference. We hope that our benchmark can pave the road for further development of multimodal video models capable of fine-grained video understanding and reasoning.

In contrast to existing video benchmarks, *TemporalBench* has the following defining characteristics:

- **Emphasis on fine-grained action understanding**. Due to the highly descriptive video captions, our negative captions highlight fine-grained temporal differences, such as *"sliced the ginger three times"* versus *"sliced the ginger twice"*, and *"put on the eyeglasses"* versus *"push the eyeglasses"*.

- **Evaluations on both short (<20 seconds) and long (>3 minute) videos.** Since the videos clips are sampled from existing videos, our benchmark can also support evaluations on long video understanding by concatenating the descriptions of multiple and non-overlapping video clips from the same source video.

- **Extends to video captioning, video grounding, and video generation.** Besides the task of video question answering, the nature of the positive captions in our benchmark allows it to seamlessly extend to evaluation of other tasks such as video temporal grounding and dense captioning.

- **Evaluations of both video embedding and question-answering models.** Given the annotated positive and negative captions in *TemporalBench*, it also supports the evaluation of discriminative

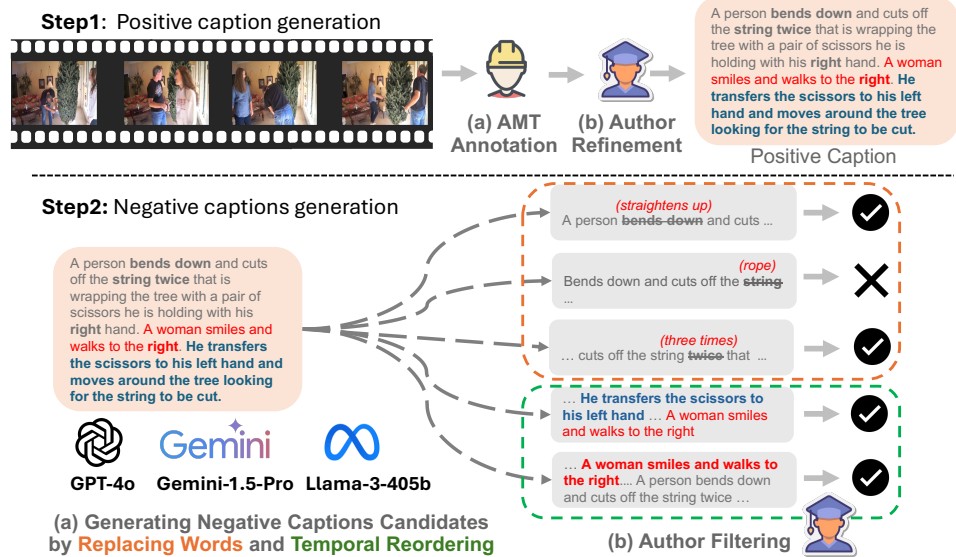

Figure 2: **Overview of the annotation pipeline for *TemporalBench*.** In step 1, we fist collect high-quality captions for the videos using qualified AMT annotators followed by refining them. In step 2, we leverage existing LLMs to generate negative captions by replacing select words and reordering the sequence of actions before filtering them ourselves.

and contrastive learning-based models such as XCLIP (Ni et al., 2022), ImageBind (Girdhar et al., 2023) as well as multimodal generative models such as GPT-4o and Gemini.

Among other observations, our empirical evaluations show that state-of-the-art multimodal video models like GPT-4o only achieve an average accuracy of 38.0% on our benchmark using our proposed multiple binary QA accuracy metric, compared to 67.9% obtained by humans. This result highlights that the aforementioned models are able to understand static visual concepts but are still limited in reasoning about the fine-grained temporal relationships of objects and events in videos. More significantly, we highlight a critical issue with using LLMs to answer multi-choice QA.

## 2 RELATED WORK

**Large Multimodal Models.** Large Language Models (LLMs) like ChatGPT (OpenAI, 2023b), GPT-4 (OpenAI, 2023c), and Llama (Touvron et al., 2023) have demonstrated impressive reasoning and generalization capabilities for text. The introduction of models that integrate visual data has brought about a significant shift in the landscape of LLMs, such as GPT-4V(ision)(OpenAI, 2023a). Building upon open-source LLMs (Touvron et al., 2023; Chiang et al., 2023), a wide range of multimodal models has achieved remarkable progress, led by pioneering models such as LLaVA (Liu et al., 2023a; 2024a) and MiniGPT-4 (Zhu et al., 2024b), which combine LLMs' capabilities with a CLIP (Radford et al., 2021) based image encoder. Recently, a growing number of LMMs have been developed to handle a wider range of tasks and modalities, such as region-level LMMs (Cai et al., 2024a; Zhang et al., 2023c; Chen et al., 2023; Peng et al., 2023; Zhang et al., 2023b), 3D LMMs (Hong et al., 2023), and video LMMs (Lin et al., 2023; Zhang et al., 2023a; 2024b).

**Multimodal Understanding Benchmarks.** The recent significant advancements have resulted in more versatile multimodal models, making it imperative to thoroughly and extensively evaluate their visual understanding and reasoning abilities. Conventional multimodal benchmarks like VQA (Antol et al., 2015), GQA (Hudson & Manning, 2019) and VizWiz (Gurari et al., 2018) have been revitalized and used for evaluating the general visual question answering performance for LMMs. Some other question answering benchmarks like TextVQA (Singh et al., 2019), DocVQA (Mathew et al., 2021) and InfoVQA (Mathew et al., 2022) have also been employed to validate the text-oriented understanding. Recent studies have introduced a variety of new benchmarks, such as SEED-Bench (Li et al., 2023a), MMBench (Liu et al., 2023b) and MM-Vet (Yu et al., 2024b) for evaluating the models' integrated problem-solving capabilities, and MMMU (Yue et al., 2024a) and MathVista (Lu et al.,

2024) for scientific and mathematical reasoning. In addition, the commonly known hallucination problem also appears in LMMs, and is also investigated in POPE (Li et al., 2023b), MMHal-Bench (Sun et al., 2023) and Object HalBench (Yu et al., 2024a), *etc*.

**Video Understanding Benchmarks.** Recently, an increasing amount of research is transitioning its focus from the image to the video domain. Videos differ from images in that they possess more complex content with temporal dynamics. This unique aspect calls for a different set of metrics and benchmarks. Many efforts have leveraged existing video question answering benchmarks (Xu et al., 2017; Yu et al., 2019b; Xiao et al., 2021) built on top of video-text datasets (Chen & Dolan, 2011; Xu et al., 2016; Zhang et al., 2019). More recently, several LMM-oriented benchmarks have been proposed for different aspects such as long-form egocentric understanding with EgoSchema (Mangalam et al., 2024), and temporal understanding and ordering like Tempcompass (Liu et al., 2024c). MV-Bench (Li et al., 2024b) compiles existing video annotations from different disciplines into a new benchmark, while Video-MME (Fu et al., 2024) and MMWorld (He et al., 2024b) claim to support a comprehensive evaluation of video understanding and world modeling, respectively. Our *TemporalBench* serves the common goal of evaluating models for video understanding but differs in several aspects. On the one hand, we exhaustively curate videos from different domains and ask human annotators to annotate the visual contents with as much detail as possible. On the other hand, we particularly focus on temporal dynamics such as human actions and human-object interactions that exist exclusively in videos and which are crucial for video understanding, reasoning and forecasting. While the ShareGPT4Video dataset (Chen et al., 2024) also contains long captions, theirs differ from ours by being entirely generated by GPT-4o instead of annotated by humans.

## 3  *TemporalBench*

Compared to static images, videos inherently contain significantly more fine-grained temporal information, as they capture the unfolding of actions and events over time. Existing multimodal video understanding benchmarks (Xu et al., 2016) mostly evaluate models' coarse-level understanding of videos. An example from the recent Seed-Bench dataset is the question, "What action is happening in the video?" with the answer, "moving something up." However, such types of coarse-level video questions have been demonstrated to be easily solved with just a single frame (Wu, 2024) or even by a text-only LLM (Tan et al., 2024; Mangalam et al., 2024).

Such phenomena arises due to a fundamental limitation in the text descriptions in those benchmarks. As a result of their coarseness, the positive and negative options for video question-answering can usually be distinguished without understanding the temporal dynamics, such as the models only needing to choose between "The man is cooking" and "The man is exercising".

To address this limitation, we carefully design a human annotation pipeline to curate highly detailed descriptions about the activities in the videos. Given the detailed video clip descriptions, such as *A right hand holds a piece of peeled ginger while a knife is held in the left and makes 3 slices off the ginger.*, the negative captions can be curated to truly reflect whether a model understands the temporal dynamics, such as changing *"three slices"* into *"two slices"*. In a nutshell, such highly detailed temporal annotations can be used to carefully examine whether a multimodel video model truly understands the temporal state transition in videos.

Our benchmark enriches several fundamental video understanding tasks due to its detailed captions:

- **Fine-grained video question answering.** Given a detailed positive caption, multimodal video models need to distinguish it from the associated negative where a slight modification is made to temporal descriptions, *e.g., "push the eyeglasses up"* versus *"pull the eyeglasses down"*, or *"cut 3 slices off"* versus *"cut 2 slices off"*.

- **Fine-grained video captioning.** Our detailed video captions can naturally enrich the video captioning task, different from current video captioning tasks such as MSRVTT (Xu et al., 2016) which focus on coarse-level descriptions.

- **Long video understanding and fine-grained activity inspection.** Since the video clips are extracted from a long source video, the respective video clip descriptions can be concatenated to form a longer video description which can be pivoted to the long video understanding task, where we find that all current multimodal video models suffer.

- **Dense video-text matching and retrieval.** Our detailed video captions can be naturally employed to evaluate video-language embedding models such as VideoCLIP (Xu et al., 2021). Given a positive caption and several negative captions, we can evaluate whether CLIP (Radford et al., 2021) based video embedding models can distinguish the subtle differences in captions. In addition, given a set of positive video-text pairs, video retrieval performance can be evaluated, similar to image retrieval on COCO (Lin et al., 2014) and Flickr30K (Young et al., 2014).

- **Video grounding from detailed text descriptions.** Since the video clips are cropped from the source video, with the documented starting and ending time, our benchmark can serve as a fine-grained moment localizing benchmark from text descriptions. This is different from existing video grounding datasets such as Charades-STA (Gao et al., 2017), COIN (Tang et al., 2019), Ego4D (Grauman et al., 2024) where the text descriptions are usually very short, possibly resulting in low temporal localization performance due to the vague and coarse descriptions.

- **Text-to-Video (T2V) generation with detailed prompts.** Given our highly detailed description, a T2V generation model can be evaluated by verifying if the generated videos reflect the fine-grained action details.

Next, we detail the dataset curation and evaluation setup for *TemporalBench*.

## 3.1 VIDEO COLLECTION

We collect video clips from a wide range of sources across diverse domains, where the majority comes from existing video grounding benchmarks. Our dataset includes a wide spectrum of video types from seven sources, including (1) procedure videos *e.g.,* COIN (Tang et al., 2019), (2) human activities *e.g.,* ActivityNet-Captions (Yu et al., 2019a) and Charades (Krishna et al., 2017), (3) ego-centric videos *e.g.,* EgoExo4D (Grauman et al., 2024), (4) movie descriptions (Rohrbach et al., 2015), (5) professional gymnasium videos *e.g.,* FineGym (Shao et al., 2020), and (6) unexpected humor videos Oops (Epstein et al., 2020). We sample around 300 video clips from the validation and test sets of each video dataset, which results in 2k videos. The statistics of *TemporalBench* is shown in Table 1.

We intentionally filter out video clips that (1) are mostly static by leveraging optical flow (Farnebäck, 2003), (2) contain multiple scene transitions by leveraging PySceneDetect [1] and (3) last longer than 20 seconds. We observe that the large amount of information in long videos make it difficult for annotators to provide detailed action descriptions. The distribution of video lengths is shown in Figure 3. Additionally, we remove the audio from the videos during annotation to ensure that all informative signals come solely from the visual frames, preventing the answers from being influenced by the audio.

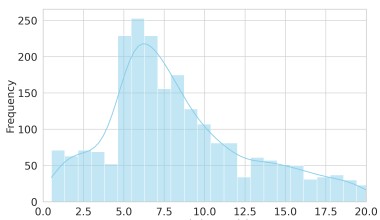

Figure 3: Video length distribution of *TemporalBench*.

## 3.2 VIDEO CAPTION ANNOTATION PROCESS

**Positive Captions Annotation.** We employ a two-stage human labeling process for curating video captions with fine-grained activity descriptions, where the qualified Amazon Mechanical Turk (AMT) workers are first instructed to give a detailed video caption. Then, the authors of this work refine the caption by correcting the mistakes and adding missing details *w.r.t.* the actions. The overall pipeline is shown in Figure 2. All video clips are annotated following the same pipeline except for Finegym (Shao et al., 2020) as it has already provided accurate and detailed action descriptions for professional gymnasium videos. Consequently, we reuse its annotations.

We first use 3 probing video captioning questions with 2 in-context examples as the onboarding task for AMT master workers. We manually inspect the soundness and amount of temporal details of the AMT worker captions to select high quality AMT video captioning workers. During the annotation process by AMT workers, we also continue to remove the unqualified workers based on the ratio of the captions that authors in this paper refined. In this way, we ensure that the AMT provides a high quality initial point for positive captions.

---

[1] https://www.scenedetect.com/

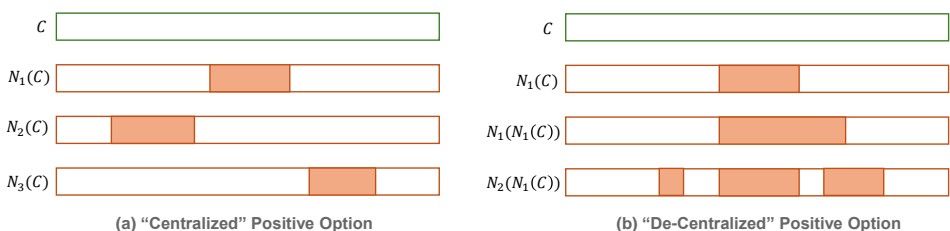

Figure 4: An illustration of multi-choice QA with (a) "centralized" and (b) "de-centralized" positive option. Orange blocks indicate the altered contents from the positive option (green box).

**Negative Caption Annotation.** Our negative captions are aimed at confusing multimodal video models with respect to fine-grained activity details, such as changing *"cut a ginger twice using a knife"* to *"cut a ginger three times using a knife"*. We construct negatives upon two granularities: word level and event level. Specifically, word level negatives denote the case where a certain word or phrase is replaced while event level negatives denote the case where the order of two events are reversed. Empirically, we find that LLMs can produce more creative and diverse negatives compared to AMT workers and authors. Therefore, we leverage three leading LLMs, GPT-4o (OpenAI, 2024), Gemini-1.5-Pro (Gemini Team, 2024) and Llama-3.1-405b (Meta, 2024) to curate a diverse set of negative caption candidates instructed by 3 in-context examples, with up to 9 negatives at word level and 6 negatives at event level.

Afterwards, the authors of this work review those negative caption candidates in the format of multi-choice QA, which results in our complete *TemporalBench* dataset with ∼2K high-quality human-annotated video captions and ∼10K video question-answer pairs.

### 3.3 A Pitfall in Multi-choice Question Answering

A conventional approach to evaluate large multimodal models is using the multi-choice question-answering format, which is adopted by the majority of current benchmarks including MMMU (Yue et al., 2024a), MathVista (Lu et al., 2024), EgoSchema (Mangalam et al., 2024) etc. However, indicated by recent studies by (Cai et al., 2024b) and (Yue et al., 2024b), a pure LLM can achieve comparable or even stronger performance on those benchmarks without looking at the visual content at all. Recent studies argue that (1) some questions are not designed well so that the question can be answered without looking at the visual content, or (2) the model memorizes the QA pairs, *i.e.,* data contamination occurs.

While developing our benchmark, we notice another previously ignored but critical pitfall for multi-choice QA. Specifically, if every negative answer choice is generated by changing a small part of the correct answer, the LLM can detect those changes to find a "centralized" description and use that cue for its prediction. To study this, given a positive caption $C$ and its associated negative caption $N(C)$, we intentionally derive a few negatives from $N_1(C)$ (instead of for $C$), resulting in $N_1(N_1(C))$ and $N_2(N_1(C))$, resulting in $[C, N_1(C), N_1(N_1(C)), N_2(N_1(C))]$ as options, so that $N_1(C)$ becomes the "centralized" description (see Fig. 4). Surprisingly, we find that 62% of text-only GPT-4o's predictions correspond to $N(C)$, while only 18% of its predictions correspond to $C$. Our findings also align with human behavior analysis from psychology (Furman & Wang, 2008), where humans can achieve better than random chance performance on multi-choice QAs using similar cues.

Motivated by this findings, we propose to decompose a single multi-choice QA into multiple binary QAs. In this case, we eliminate the "centralized option" due to the fact that there are only two options to choose from. As a result, given $M$ negatives, the multiple binary QAs will query a model $M$ times, where the random chance performance changes from $\frac{1}{M+1}$ to $(\frac{1}{2})^M$. Given that $(\frac{1}{2})^M > \frac{1}{M+1}$ for every $M > 2$, multiple binary QA is a more difficult task than multi-choice QA.

## 4 Experiments

### 4.1 Experiment Setup

We evaluate both (1) multimodal video text generation models, including GPT-4o (OpenAI, 2024), Gemini-1.5-Pro (Gemini Team, 2024), Claude-3.5-Sonnet (Anthropic, 2024), Qwen2VL (Wang et al., 2024), LLaVA-OneVision (Li et al., 2024a), LLaVA-Next-Video (Zhang et al., 2024b), Phi-

3.5-Vision (Abdin et al., 2024), MiniCPM-2.6 (Yao et al., 2024), MA-LMM (He et al., 2024a), VideoLLaVA (Lin et al., 2023), InternLM-Xcomposer-2.5 (Zhang et al., 2024a), and (2) multimodal video embedding models, including XCLIP (Ni et al., 2022), ImageBind (Girdhar et al., 2023), and LanguageBind (Zhu et al., 2024a). We exponentially increase the number of frames to study its effect on video understanding. More details can be found in Appendix C.

To study the effect of single frame bias and text bias, we also evaluate models trained on single images, including LLaVA-1.5 (Liu et al., 2024a), LLaVA-NeXT (Liu et al., 2024b), and Phi-3V (Abdin et al., 2024). In the latter case, we evaluate the LLMs including GPT-4o (OpenAI, 2024), Gemini-1.5-Pro (Gemini Team, 2024), Yi-34B (Young et al., 2024), Vicuna (Chiang et al., 2023) and Flan-T5 (Wei et al., 2021) without using videos at all.

## 4.2 HUMAN PERFORMANCE

We use Amazon Mechanical Turk to evaluate human performance. Note that we exclude the positive caption annotators to ensure that there is no data contamination. Again, we use an onboarding test using a held out binary video QA evaluation set which has clear answers. Next, we show the performance on each task.

## 4.3 FINE-GRAINED VIDEO QUESTION ANSWERING

The results for multimodal generative models and embedding models are shown in Table 2. Several interesting findings arise:

**The performance of any video model is far from human performance.** As shown in the table, humans show an average performance of 67.9%, which is significantly higher than the best models, GPT-4o and Qwen2VL-72B, by ∼30%. Therefore, there is a large gap between model's performance and human performance. Note that we are employing standard AMT workers instead of domain experts, meaning that the expert-level accuracy can be even higher, especially for professional video understanding like FineGym.

**Models show limited performance gains with more frames**. As shown in Figure 5, with more frames, multimodal video models usually show better performance. However, performance generally saturates around 8-16 frames, meaning that models struggle to improve fine-grained activity understanding even with more frames. This is a clear contrast with human performance, showing that there is still a large space for multimodal video models to improve.

**Multiple Binary QA is a more challenging metric.** Multiple Binary QA, as proposed in Section 3.3, prevents a model from exploiting cues in the answer choices, and evaluates whether a model truly understands the temporal dynamics in the video by splitting a single $M + 1$-way multiple choice question into $M$ binary choice questions. For example, GPT-4o receives 76.0% accuracy but only 38.0% on multiple binary accuracy, showing a huge gap. These results indicate that understanding the fine-grained temporal dynamics is still a challenging task for current proprietary models and open-sourced models.

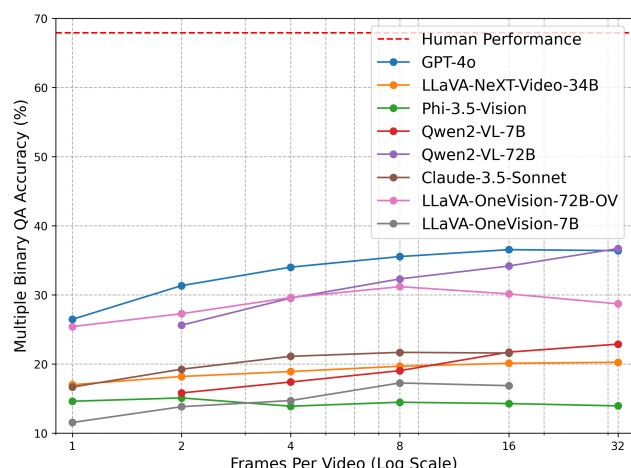

Figure 5: **Model performance on *TemporalBench*** with varying frames.

**Video Embedding models show near chance performance.** All multimodal video embedding models, including XCLIP, LanguageBind, and ImageBind show near random chance performance. One reason could be that their small embedding

Table 1: Dataset characteristics including number of samples, average length, single image bias, and language bias.

| Dataset | Number of Samples | Org. Avg. # words | Ours Avg. # words |
|---|---|---|---|
| ActivityNet (Krishna et al., 2017) | 281 | 13.03 | 49.55 |
| EgoExo4d (Grauman et al., 2024) | 307 | 7.73 | 47.79 |
| Charades (Gao et al., 2017) | 298 | 6.21 | 44.16 |
| MPI Movie Description (Rohrbach et al., 2015) | 326 | 12.39 | 35.33 |
| Oops (Epstein et al., 2020) | 294 | 10.06 | 43.27 |
| COIN (Tang et al., 2019) | 385 | 5.01 | 50.06 |
| FineGym (Shao et al., 2020) | 288 | 21.92 | 21.92 |
| *TemporalBench* (ours) | 2179 | 10.9 | 41.72 |

Table 2: *TemporalBench* performance of various multimodal generative models and embedding models under the binary QA accuracy (BA) and multiple binary QA settings (MBA). The prefix "T-" indicates the annotated subset in our *TemporalBench*.

| Model | T-ActivityNet | T-Charades | T-FineGym | T-Movie | T-Oops | T-COIN | T-EgoExo | BA | MBA |
|---|---|---|---|---|---|---|---|---|---|
| Human Performance | **69.4** | **81.9** | **35.8** | **74.5** | **69.7** | **70.6** | **70.7** | **89.7** | **67.9** |
| Random Chance | 11.0 | 13.7 | 6.1 | 12.0 | 5.6 | 11.1 | 5.6 | 50.0 | 9.4 |
| *Video Embedding Models: Text + Multiple Frames as Input* | | | | | | | | | |
| XCLIP | 14.2 | 16.1 | 7.3 | 19.9 | 8.8 | 15.2 | 6.8 | 51.6 | 12.8 |
| ImageBind | 17.4 | 16.8 | 7.3 | 19.0 | 11.2 | 16.1 | 9.1 | 53.0 | 14.0 |
| LanguageBind | 22.4 | 15.1 | 6.3 | 19.3 | 10.9 | 15.6 | 11.1 | 52.8 | 14.5 |
| *Video Multimodal Generative Models : Text + Multiple Frames as Input* | | | | | | | | | |
| GPT-4o | 48.8 | 42.6 | 16.7 | 43.9 | 34.4 | 42.9 | 34.5 | 76.0 | 38.0 |
| Gemini-1.5-Pro | 34.9 | 24.5 | 8.0 | 35.6 | 22.8 | 34.0 | 21.8 | 67.4 | 26.4 |
| Claude-3.5-Sonnet | 29.5 | 27.5 | 13.2 | 29.1 | 14.6 | 27.8 | 21.2 | 65.9 | 23.5 |
| InternLM-XC2.5 | 25.3 | 34.9 | 19.4 | 38.7 | 25.9 | 18.2 | 16.6 | 58.7 | 25.2 |
| VideoLLaVA | 34.9 | 29.2 | 13.5 | 25.5 | 20.7 | 32.5 | 20.2 | 67.2 | 25.5 |
| MA-LMM | 12.1 | 16.8 | 3.1 | 11.7 | 4.8 | 11.9 | 4.9 | 48.0 | 9.3 |
| Phi-3.5-Vision | 24.9 | 20.1 | 5.2 | 22.7 | 12.2 | 18.2 | 13.7 | 58.0 | 16.8 |
| MiniCPM-V2.6 | 33.1 | 25.8 | 7.6 | 29.1 | 13.6 | 22.9 | 16.0 | 62.2 | 21.3 |
| LLaVA-NeXT-Video-7B | 33.5 | 32.6 | 10.8 | 28.2 | 17.3 | 22.6 | 19.9 | 65.1 | 23.5 |
| LLaVA-NeXT-Video-34B | 30.6 | 26.8 | 10.4 | 24.8 | 18.0 | 24.9 | 17.3 | 64.0 | 22.0 |
| LLaVA-OneVision-7B | 30.2 | 27.5 | 7.6 | 25.8 | 16.0 | 22.1 | 14.3 | 60.0 | 19.7 |
| LLaVA-OneVision-72B | 43.8 | 34.2 | 11.5 | 35.3 | 27.9 | 33.0 | 28.3 | 70.5 | 30.7 |
| Qwen2-VL-7B-Instruct | 32.4 | 31.9 | 4.5 | 35.9 | 18.4 | 25.2 | 21.8 | 64.6 | 24.9 |
| Qwen2-VL-72B-Instruct | 43.4 | 42.6 | 16.7 | 45.1 | 36.4 | 43.4 | 37.1 | 75.8 | 38.2 |
| *Large Multimodal Models (LMMs): Text + 1 Frame as Input* | | | | | | | | | |
| GPT-4o | 32.0 | 30.2 | 15.3 | 31.0 | 26.5 | 33.8 | 27.7 | 70.0 | 28.4 |
| LLaVA-1.5-13B | 16.0 | 17.1 | 9.4 | 16.6 | 6.1 | 16.1 | 9.1 | 55.6 | 13.1 |
| LLaVA-1.5-7B | 25.3 | 25.8 | 8.7 | 19.3 | 9.2 | 22.1 | 16.6 | 60.5 | 18.3 |
| Phi-3-Vision-128k-Instruct | 22.8 | 19.8 | 4.5 | 17.8 | 8.5 | 17.7 | 14.7 | 54.4 | 15.3 |
| *Large Larguage Models (LLMs): Text as Input* | | | | | | | | | |
| GPT-4o | 30.2 | 31.9 | 16.7 | 27.9 | 22.8 | 27.5 | 28.0 | 67.7 | 26.5 |
| Gemini-1.5-Pro | 22.4 | 20.5 | 4.5 | 19.9 | 10.2 | 16.6 | 17.9 | 58.0 | 16.0 |
| Yi-34B | 20.6 | 27.5 | 10.4 | 21.8 | 11.2 | 23.4 | 16.9 | 59.9 | 18.3 |
| Vicuna7b-1-5 | 19.2 | 17.4 | 6.6 | 11.0 | 5.1 | 12.5 | 7.8 | 50.4 | 9.8 |
| Flan-T5-XL | 24.6 | 23.5 | 5.6 | 19.9 | 11.9 | 23.1 | 14.0 | 57.8 | 17.8 |

size (typically a vector with size around 768-2048) is insufficient to capture fine-grained temporal details.

**Low single-frame bias and language bias.** As shown in Figure 5 and Table 6, the performance of models like GPT-4o gradually increases with more frames. Excluding GPT-4o, all remaining VLMs are trained with single images *e.g.,* LLaVA-1.5, Phi-3V, and text-only LLMs such as Yi-34B and Vicuna-7B.

## 4.4 VIDEO CAPTIONING

Our detailed video captions also enables analyzing a model's fine-grained video captioning capabilities. For this, we prompt multimodal video models to generate a caption for an input video, with 3 captioning examples in the prompt as guidance to mimic the style of our detailed video captions. We evaluate the resulting video captioning performance using classical image captioning metrics,

Table 3: Comparison of models for video captioning using Caption Similarity, CIDEr, BLEU, and ROUGE metrics. Cosine similarity using sentence transformer reflects the captioning quality the best.

| Model | Similarity | CIDEr | ROUGE | BLEU_1 | BLEU_2 | BLEU_3 | BLEU_4 |
|---|---|---|---|---|---|---|---|
| **Video Multimodal Generative Models : Text + Multiple Frames as Input** | | | | | | | |
| GPT-4o | **63.47** | 6.59 | **19.99** | 23.70 | **11.74** | **5.90** | **3.09** |
| Gemini-1.5-Pro | 56.54 | **10.98** | 19.11 | 18.96 | 9.19 | 4.53 | 2.36 |
| Claude-3.5-Sonnet | 54.13 | 8.64 | 17.14 | 24.35 | 10.32 | 4.43 | 2.05 |
| VideoLLaVA | 45.97 | 4.49 | 16.95 | 12.59 | 5.44 | 2.29 | 1.03 |
| MA-LMM | 38.72 | 3.07 | 14.99 | 10.09 | 4.81 | 2.24 | 1.06 |
| Phi-3.5-Vision | 42.93 | 3.67 | 16.54 | 20.36 | 8.38 | 3.40 | 1.58 |
| MiniCPM | 47.24 | 1.50 | 14.18 | 15.53 | 5.45 | 1.92 | 0.79 |
| LLaVA-NeXT-Video-7B | 50.09 | 2.31 | 15.84 | 18.07 | 6.98 | 2.60 | 1.05 |
| LLaVA-NeXT-Video-34B | 53.13 | 5.33 | 15.92 | 21.43 | 9.17 | 4.02 | 1.83 |
| LLaVA-OneVision-7B | 50.33 | 1.43 | 16.08 | 16.17 | 6.99 | 2.92 | 1.33 |
| LLaVA-OneVision-72B | 53.90 | 8.00 | 18.23 | 22.08 | 10.63 | 5.31 | 2.78 |
| Qwen2-VL-7B-Instruct | 51.93 | 6.87 | 18.03 | 12.45 | 6.07 | 3.00 | 1.56 |
| Qwen2-VL-72B-Instruct | 56.13 | 9.31 | 19.11 | 15.71 | 8.03 | 4.14 | 2.24 |
| **Large Multimodal Models (LMMs): Text + 1 Frame as Input** | | | | | | | |
| GPT-4o | 52.32 | 7.29 | 17.10 | **25.07** | 11.09 | 5.04 | 2.41 |
| LLaVA-1.5-13B-HF | 47.92 | 4.90 | 18.04 | 22.62 | 9.78 | 4.23 | 2.03 |
| LLaVA-1.5-7B-HF | 45.68 | 6.87 | 17.82 | 21.95 | 9.53 | 4.17 | 1.98 |
| Phi-3-Vision-128k-Instruct | 41.96 | 4.00 | 16.10 | 19.86 | 8.29 | 3.42 | 1.59 |

CIDEr (Vedantam et al., 2015), BLEU (Papineni et al., 2002) at different n-gram levels, ROUGE (Lin, 2004), as well as the embedding similarity with sentence transformer (Reimers & Gurevych, 2019) between the ground truth caption and the generated caption.

Results in Table 3 show that GPT-4o achieves the best performance. Interestingly, the results indicate that the embedding similarity aligns most closely with the video QA task results from Sec 4.3. Other classical captioning metrics show inconsistent results. For example, GPT-4o obtains better performance with one compared to 64 frames on both CIDEr and BLEU scores (e.g., for CIDEr 7.29 vs. 6.59). On the other hand, all models show similar ROUGE scores. Thus, for the zero-shot captioning task, our findings indicate that text embedding similarity may be the most reliable metric.

### 4.5 Long Video Understanding

Since our benchmark is annotated at the video clip level, we can easily extend it to long video understanding by concatenating the captions of different video clips within the same original video. In our study, we choose video datasets whose original length is both short (AcitivityNet and Charades, average length < 3 minutes) and long (COIN and FineGym, > 20 minutes). We randomly sample video clips within the same original video, and then crop a new video segment whose starting time corresponds to that of the earliest sampled video clip and whose ending time corresponds to that of the latest sampled video clip. We then concatenate all the sampled video captions together to form a single long detailed description corresponding to the new video segment. Given this positive caption, we generate negative captions for it by replacing the positive caption of one of the sampled video clips with its negatives. The model is then tasked to choose the correct long caption out of multiple choices. We set the number of negative options to be ∼4, resulting in a similar random chance performance as in Sec 4.3. In this way, we investigate whether multimodal video models can understand and distinguish fine details in a long video.

We show in Table 7 (supplemental), that all multimodal video models show a significant performance drop for this task compared to short video understanding. This is also reflected in all models performing better on relatively shorter videos (ActivityNet and Charades) compared to longer videos (COIN and FineGym). These results indicate that finding the subtle temporal dynamic differences in a long video is indeed an extremely difficult task. It is similar in nature to the needle-in-the-sea task (Kamradt, 2023) in NLP except in the temporal domain. We hope that *TemporalBench* for long video understanding can serve as a very challenging task for future video understanding model development.

Table 5: *TemporalBench* statistics on each category on binary QA accuracy.

| Action Order | Action Frequency | Action Type | Motion Magnitude | Motion Direction | Action Effector | Event Reorder | Others | Overall |
|---|---|---|---|---|---|---|---|---|
| 130 | 531 | 2812 | 321 | 1554 | 1118 | 2105 | 1347 | 9918 |

## 5 IN-DEPTH ANALYSIS

### 5.1 WHY MULTIPLE BINARY QA INSTEAD OF MULTI-CHOICE QA?

As discussed in Section 3.3, in the standard multi-choice QA setting, if negatives are all slightly variations of the positive caption, we find that LLMs can determine the "centralized" caption, and take a shortcut to achieve better performance. To demonstrate this, based on one negative caption $N(C)$ in *TemporalBench*, we intentionally generate two negative captions derived from $N(C)$ (instead of $C$), resulting in $N_1(N(C))$ and $N_2(N(C))$. Given two set of options $[C, N_1(C), N_2(C)), N_3(C))]$ and $[C, N_1(C), N_1(N_1(C)), N_2(N_1(C))]$ shown in Figure 4, text-only GPT-4o displays different behaviors. As shown in Table 4, under the intentionally designed negative options, GPT-4o will choose $N_1(C)$ with 66.4% probability. This again demonstrates the necessity and advantage of our multiple binary QA accuracy (MBA) metric design over the standard multi-choice QA setting.

### 5.2 PERFORMANCE ON CATEGORIES

Broadly, *TemporalBench* evaluates word level replacement and event level re-ordering. Here we further breakdown the word level replacement into following categories: 1. Action order (change the order); 2. Action frequency (1 times v.s. two times); 3. Action type (put vs pull); 4.

Table 4: Effect of the "Centralized" Caption on text-only GPT-4o.

| Percentage of Predictions Aligned with -> | $C$ | $N_1(C)$ |
|---|---|---|
| "Centralized" Negative | 83.3 | 6.4 |
| "De-Centralized" Negatives | 17.7 | 66.4 |

Motion magnitude (slightly vs intensively); 5, Motion Direction/Orientation (forward vs backward, circular vs back-and-forth). 6. Action effector (cutting with left hand vs cutting with right hand) 7. Others. We prompt GPT-4o to perform 7-way classification and show the per-category performance in Table 8 (supplemental). Results indicate that multimodal video models shows better performance on "others" category rather than the other categories related to actions. Among the seven categories, models struggle most on action frequency (counting), which show that they do not memorize repeated occurrences well.

## 6 CONCLUSION AND FUTURE WORK

We propose TemporalBench, a novel video understanding benchmark, to evaluate the fine-grained temporal understanding abilities of multimodal video models. The video captions in our benchmark are significantly denser than existing datasets such as MSRVTT and TGIF, offering detailed temporal annotations. TemporalBench also provides a more challenging set of tasks that push current multimodal models beyond coarse-level understanding. The empirical results reveal a substantial gap between human performance and current state-of-the-art models. We hope that this benchmark fosters further research in developing models with enhanced temporal reasoning capabilities. Our benchmark could also be easily utilized for other fundamental video tasks such as spatio-temporal localization and text-to-video generation with fine-grained prompts.

**Limitations.** One cannot fully analyze the behavior of proprietary models included in this paper due to the lack of access to these models, which are GPT-4o, Gemini-1.5-Pro and Claude 3.5 Sonnet.

## REPRODUCIBILITY STATEMENT

We attach part of the dataset in the submission's supplementary materials. We will also publicly release it along with the code used to evaluate the LMMs upon the paper's acceptance.

## ETHICS STATEMENT

This research primarily utilizes publicly available video datasets, which have been collected and annotated by qualified annotators and authors, ensuring compliance with ethical standards. We have made every effort to ensure that the data used respects privacy and contains no personally identifiable information. Furthermore, we acknowledge the potential implications of fine-grained video understanding, especially in sensitive applications such as surveillance and autonomous systems. As such, we advocate for responsible and ethical use of this research, urging caution in deploying these models in real-world scenarios to avoid harmful or unintended consequences.

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

## A  BROADER IMPACT

*TemporalBench*, a comprehensive benchmark for video understanding, has the potential to significantly advance research in this field by offering improved metrics for model evaluation. Our work aims to enhance the temporal reasoning capabilities of future video understanding models. However, the broader impact of more advanced video understanding technologies raises important societal concerns, including the risk of mass surveillance, privacy violations, and the development of harmful applications like autonomous weapons. Therefore, we strongly encourage thoughtful consideration when deploying these models in real-world scenarios to mitigate negative or unintended consequences.

## B  MORE VISUALIZATIONS OF OUR BENCHMARK

In this section, we present comprehensive visualizations of our fine-grained annotations with both positive and negative descriptions. For each benchmark mentioned in Table 1, we provide one video example with its positive annotation and one of the corresponding negative descriptions (there are more than one negative for a single video in our dataset) in Figures 6 & 7. The video examples (*a - f*) are displayed in the same order as their sources in Table 1 (7 in total).

## C  MORE RESULTS WITH EXTENDED FRAMES

In the main paper, we only report the performance of each multimodal video models with the the number of frams that leads to the best performance. Here we extend the results to show the results of more frames in Table 6.

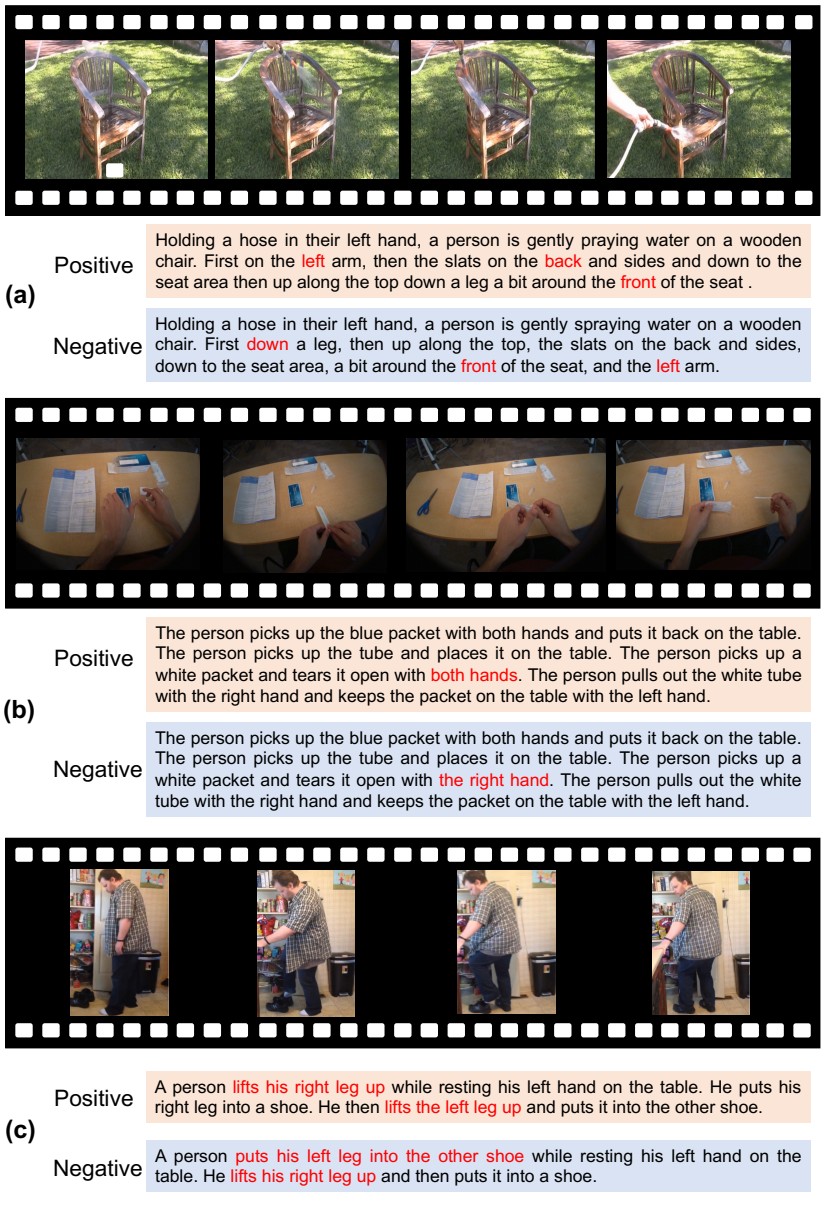

Figure 6: Visualizations (I) of our fine-grained annotations of the videos with both positive and negative descriptions.

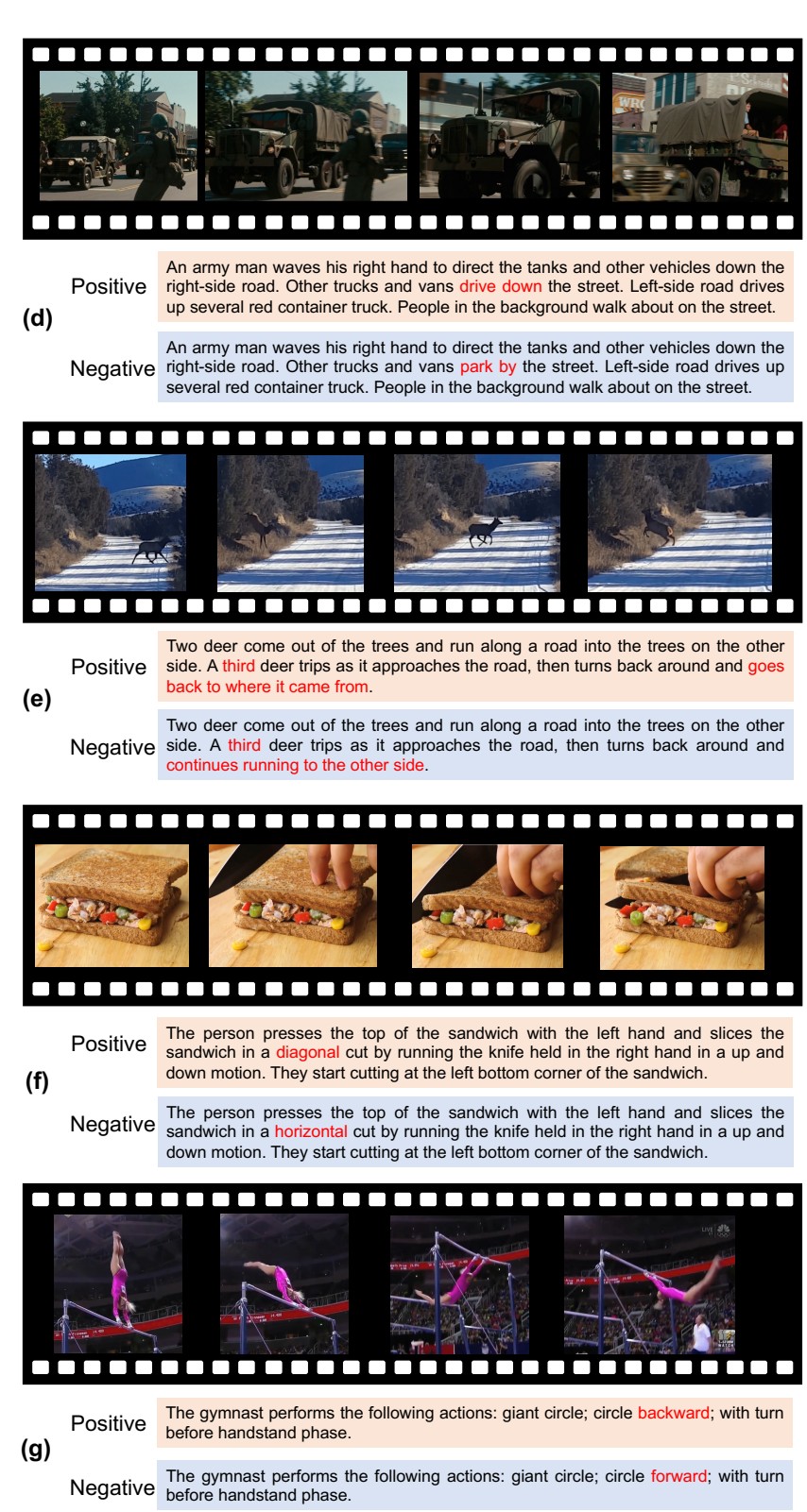

Figure 7: Visualizations (II) of our fine-grained annotations of the videos with both positive and negative descriptions.

Table 6: *TemporalBench* performance of various models under binary QA and multiple binary QA setting.

| Model | Frames Per Video | Multiple Binary Accuracy (%) | Binary QA Accuracy (%) |
|---|---|---|---|
| Human | - | **67.9** | **89.7** |
| Random Chance | - | 9.4 | 50.0 |
| XCLIP | 8 | 12.8 | 51.6 |
| ImageBind | 2 | 14.0 | 53.0 |
| LanguageBind | 8 | 14.5 | 52.8 |
| GPT-4o | 64 | 38.0 | 76.0 |
| | 32 | 38.2 | 75.9 |
| | 16 | 38.4 | 75.7 |
| | 8 | 37.3 | 75.1 |
| | 4 | 35.8 | 74.4 |
| | 2 | 33.2 | 72.7 |
| | 1 | 28.4 | 70.0 |
| | 0 | 26.5 | 67.7 |
| Gemini-1.5-Pro | 1fps | 26.4 | 67.4 |
| | 0 | 16.0 | 58.0 |
| Claude-3.5-Sonnet | 16 | 23.5 | 65.9 |
| | 8 | 23.6 | 65.4 |
| | 4 | 23.0 | 64.7 |
| | 2 | 21.2 | 61.8 |
| | 1 | 18.4 | 58.4 |
| InternLM-XC25 | 1fps | 25.2 | 58.7 |
| LLaVA-NeXT-Video-34B-DPO | 32 | 22.0 | 64.0 |
| | 16 | 21.8 | 63.7 |
| | 8 | 21.4 | 63.3 |
| | 4 | 20.7 | 63.0 |
| | 2 | 19.9 | 61.9 |
| | 1 | 18.8 | 60.5 |
| LLaVA-NeXT-Video-7B-DPO | 32 | 17.2 | 59.6 |
| | 16 | 22.3 | 64.0 |
| | 8 | 23.5 | 65.1 |
| | 4 | 22.9 | 64.2 |
| | 2 | 21.4 | 63.1 |
| | 1 | 19.0 | 62.0 |
| VideoLLaVA | 8 | 25.5 | 67.2 |
| Phi-3.5-Vision-Instruct | 15.5 | 56.7 | |
| | 16 | 15.9 | 57.2 |
| | 8 | 15.9 | 57.4 |
| | 4 | 15.5 | 57.5 |
| | 2 | 16.8 | 58.0 |
| | 1 | 16.4 | 57.8 |
| Qwen2-VL-7B-Instruct | 32 | 24.9 | 64.6 |
| | 16 | 23.5 | 63.2 |
| | 8 | 20.9 | 60.9 |
| | 4 | 19.2 | 59.5 |
| | 2 | 17.6 | 57.8 |
| Qwen2-VL-72B-Instruct | 38.2 | 75.8 | |
| | 35.5 | 74.4 | |
| | 33.8 | 73.0 | |
| | 31.0 | 71.4 | |
| | 27.3 | 69.1 | |
| MiniCPM-V-2.6 | 64 | 21.3 | 62.2 |
| LLaVA-1.5-13B-HF | 1 | 13.1 | 55.6 |
| LLaVA-1.5-7B-HF | 1 | 18.3 | 60.5 |
| Phi-3-Vision-128k-Instruct | 1 | 15.3 | 54.4 |
| Vicuna7B-1.5 | 0 | 9.8 | 50.4 |
| Yi34BNousYi | 0 | 18.3 | 59.9 |
| FastChat-FlanT5 | 0 | 11.9 | 52.2 |
| Flan-T5-XL | 0 | 17.8 | 57.8 |

Table 7: *TemporalBench* performance of various multimodal generative models and embedding models under long video understanding with multiple binary QA accuracy (MBA).

| Model | ActivityNet | Charades | FineGym | COIN | MBA |
|---|---|---|---|---|---|
| **Video Embedding Models: Text + Multi Frame as Input** | | | | | |
| XCLIP | 2.99 | 5.34 | 1.87 | 2.92 | 3.27 |
| ImageBind | 2.69 | 3.40 | 4.27 | 3.50 | 3.40 |
| LanguageBind | 4.78 | 6.31 | 3.79 | 2.72 | 4.00 |
| **Video Multimodal Generative Models : Text + Multi Frame as Input** | | | | | |
| GPT-4o | 20.30 | 21.36 | 9.81 | 17.12 | **17.12** |
| Gemini-1.5-Pro | 15.52 | 9.71 | 8.66 | 16.54 | 14.58 |
| Claude-3.5-Sonnet | 19.10 | 10.68 | 4.78 | 5.64 | 9.70 |
| VideoLLaVA | 8.96 | 6.80 | 5.07 | 2.14 | 5.29 |
| MA-LMM | 7.76 | 6.80 | 3.28 | 8.37 | 5.60 |
| Phi-3.5-Vision | 8.06 | 2.43 | 6.57 | 3.50 | 5.23 |
| MiniCPM | 8.36 | 6.80 | 3.88 | 9.53 | 9.97 |
| LLaVA-NeXT-Video-7B | 10.45 | 8.74 | 2.69 | 7.39 | 8.00 |
| LLaVA-NeXT-Video-34B | 10.75 | 10.68 | 4.78 | 3.11 | 6.90 |
| LLaVA-OneVision-7B | 8.66 | 7.77 | 5.07 | 8.56 | 8.52 |
| LLaVA-OneVision-72B | 14.93 | 10.19 | 4.18 | 5.25 | 8.65 |
| Qwen2-VL-72B-Instruct | 14.33 | 10.68 | 11.04 | 14.40 | 14.56 |
| **Large Multimodal Models (LMMs): Text + 1 frame as Input** | | | | | |
| GPT-4o | 10.45 | 12.62 | 8.33 | 11.67 | 10.80 |
| LLaVA-1.5-13B-HF | 6.57 | 5.34 | 3.88 | 3.89 | 4.84 |
| LLaVA-1.5-7B-HF | 4.78 | 5.34 | 2.69 | 3.89 | 4.01 |
| Phi-3-Vision-128k-Instruct | 8.36 | 4.85 | 3.58 | 4.67 | 5.50 |
| **Large Larguage Models (LLMs): Text as Input** | | | | | |
| GPT-4o | 11.64 | 16.99 | 7.16 | 10.70 | 11.01 |
| Gemini-1.5-Pro | 11.64 | 8.74 | 2.99 | 7.98 | 7.77 |
| Yi-34B | 7.16 | 7.28 | 5.37 | 6.61 | 6.55 |
| Vicuna7b-1-5 | 1.19 | 4.85 | 1.49 | 3.70 | 2.73 |
| Flan-T5-XL | 12.24 | 7.28 | 7.46 | 7.39 | 8.56 |

Table 8: *TemporalBench* performance under each category.

| Model | Action Order | Action frequency | Action Type | Motion Magnitude | Motion Direction | Action Effector | Event Reorder | Others | Average |
|---|---|---|---|---|---|---|---|---|---|
| **Video Embedding Models: Text + Multi Frame as Input** | | | | | | | | | |
| XCLIP | 46.2 | 50.8 | 50.9 | 56.9 | 51.2 | 51.6 | 50.2 | 55.5 | 51.6 |
| ImageBind | 43.8 | 44.8 | 55.4 | 51.1 | 52.5 | 50.4 | 48.6 | 62.0 | 53.0 |
| LanguageBind | 43.8 | 41.6 | 53.3 | 54.8 | 51.5 | 46.4 | 51.1 | 66.0 | 52.8 |
| **Video Multimodal Generative Models : Text + Multi Frame as Input** | | | | | | | | | |
| GPT-4o | 70.0 | 65.2 | **80.8** | **78.8** | 68.9 | 67.0 | 75.1 | **87.3** | **76.0** |
| Gemini-1.5-Pro | 66.9 | 60.1 | 70.8 | 70.7 | 58.6 | 59.5 | 67.7 | 79.0 | 67.4 |
| Claude-3.5-Sonnet | 63.8 | 58.0 | 71.1 | 68.2 | 60.0 | 57.4 | 62.5 | 76.7 | 65.9 |
| InternLM-XC2.5 | 53.8 | 42.4 | 61.2 | 61.4 | 52.4 | 52.4 | 59.3 | 68.3 | 58.7 |
| VideoLLaVA | 70.0 | **70.2** | 71.4 | 70.1 | **70.7** | **70.3** | 50.5 | 75.6 | 67.2 |
| MA-LMM | 54.6 | 42.7 | 48.7 | 48.9 | 46.2 | 49.4 | 49.1 | 50.8 | 48.0 |
| Phi-3.5-Vision | 53.8 | 55.4 | 60.0 | 56.1 | 53.9 | 52.2 | 55.3 | 69.4 | 58.0 |
| MiniCPM | 58.5 | 52.4 | 65.6 | 62.3 | 54.1 | 53.2 | 63.3 | 74.7 | 62.2 |
| LLaVA-NeXT-Video-7B | 68.5 | 65.5 | 68.1 | 62.0 | 66.6 | 68.7 | 52.3 | 74.2 | 65.1 |
| LLaVA-NeXT-Video-34B | 60.8 | 56.1 | 66.4 | 61.7 | 58.4 | 59.5 | 63.3 | 74.3 | 64.0 |
| LLaVA-OneVision-7B | 60.8 | 44.6 | 61.4 | 53.0 | 50.1 | 48.2 | 66.0 | 74.9 | 59.8 |
| LLaVA-OneVision-72B | 68.5 | 53.7 | 74.6 | 67.9 | 63.7 | 62.0 | 71.2 | 83.0 | 70.5 |
| Qwen2-VL-7B-Instruct | 65.4 | 46.1 | 67.3 | 66.0 | 54.5 | 54.9 | 69.3 | 75.3 | 64.6 |
| Qwen2-VL-72B-Instruct | **72.3** | 69.3 | 80.0 | **78.8** | 65.9 | 69.4 | **75.9** | 85.5 | 75.8 |
| **Large Multimodal Models (LMMs): Text + 1 frame as Input** | | | | | | | | | |
| GPT-4o | 67.7 | 65.2 | 74.0 | 70.4 | 64.3 | 62.7 | 68.6 | 78.5 | 70.0 |
| LLaVA-1.5-13B-HF | 56.9 | 52.0 | 57.6 | 53.6 | 50.3 | 53.9 | 54.2 | 63.2 | 55.6 |
| LLaVA-1.5-7B-HF | 61.5 | 61.4 | 62.1 | 54.2 | 61.6 | 65.0 | 51.1 | 67.9 | 60.5 |
| Phi-3-Vision-128k-Instruct | 46.2 | 46.3 | 56.2 | 55.8 | 48.8 | 49.6 | 56.9 | 62.3 | 54.4 |
| **Large Larguage Models (LLMs): Text as Input** | | | | | | | | | |
| GPT-4o | 64.6 | 59.9 | 73.7 | 70.1 | 61.5 | 60.2 | 69.3 | 68.7 | 67.7 |
| Gemini-1.5-Pro | 53.8 | 42.4 | 60.3 | 62.3 | 53.5 | 53.2 | 64.8 | 57.4 | 58.0 |
| Yi-34B | 53.1 | 63.1 | 59.9 | 60.4 | 56.7 | 54.8 | 65.2 | 59.3 | 59.9 |
| Vicuna7b-1-5 | 56.2 | 47.3 | 52.9 | 50.5 | 50.3 | 48.6 | 49.9 | 53.5 | 50.4 |
| Flan-T5-XL | 53.1 | 57.8 | 60.1 | 59.8 | 56.0 | 56.7 | 54.9 | 60.5 | 57.8 |

