# OpenReview forum: "TemporalBench: Towards Fine-grained Temporal Understanding for  Multimodal Video  Models"
_ICLR.cc/2025/Conference — ICLR 2025 Conference Withdrawn Submission_

### Official Review · Reviewer_h1cN · 2024-10-30

**Soundness:** 2
**Presentation:** 3
**Contribution:** 2
**Rating:** 3
**Confidence:** 4

**Summary:**

This work introduces a benchmark named TemporalBench, designed to evaluate the fine-grained temporal understanding capabilities of vision-language models (VLMs). TemporalBench is developed in two phases. First, human annotators create video captions rich in detailed temporal information, including aspects such as order, frequency, and direction, etc. Next, large language models (LLMs) are prompted to generate negative captions by altering specific elements of the human-annotated captions. VLMs are then tasked with selecting the correct caption from a pool that includes both positive and negative captions, structured as a multiple-choice video QA format. To address the issue of "centralized options", which arises from the method of generating negative captions, the authors propose using *multiple binary QAs*, where each negative caption is compared with a positive one independently. Additionally, TemporalBench can be adapted to assess long video QA and video captioning. Empirical results indicate a significant performance gap between current state-of-the-art VLMs and human capabilities on TemporalBench.

**Strengths:**

- The data collection process is well-designed, taking advantage of the accuracy of human-annotated detailed video captions alongside the efficiency and diversity of LLM-generated negative captions.
- The proposed *multiple binary QA* evaluation protocol is well-motivated and reasonable and effectively addresses the "centralized options" issue, both theoretically and empirically.
- An extensive evaluation is conducted, encompassing three embedding-based VLMs and fourteen generative VLMs, while also exploring the impact of the number of input frames.

**Weaknesses:**

- This work lacks sufficient innovation compared to existing benchmarks for video temporal understanding, and the claimed novelties are questionable:
  - **Emphasizes on fine-grained action understanding**: Most existing benchmarks, such as [1,2,3,4], already consider fine-grained action.
  - **Evaluations on both short (<20 seconds) and long (>3 minute) videos**: Video-MME [5] similarly includes videos categorized as short, medium, and long.
  - **Extends to video captioning, video grounding, and video generation**: Captioning is already evaluated in benchmarks like TempCompass [1] and Video-ChatGPT [6], while this paper does not actually investigate extensions to video grounding and generation.
  - **Evaluations of both video embedding and question-answering models**: Previous work, including Vitatecs [2] and ViLMA [4], also assesses both video embedding and generative VLMs.
- Some results indicate potential quality issues with the data in TemporalBench:
  - Human performance on TemporalBench is merely 67.9%.
  - GPT-4o, when given only text input, achieves 67.7% accuracy on binary QA, suggesting that a significant portion of negative captions can be identified without reference to the video.
- The evaluation of detailed video captioning in Section 4.4 relies on classical image captioning metrics like CIDEr, BLEU, and ROUGE, which may not be suitable for assessing detailed video captioning.
- Simply concatenating the captions from short video clips to create long video captions can lead to ambiguity; for example, the caption for the first clip may contradict the content of the second clip. This inconsistency can undermine the reliability of the long video QA results.

[1] TempCompass: Do video LLMs really understand videos?

[2] Vitatecs: A diagnostic dataset for temporal concept understanding of video-language models.

[3] MVBench: A Comprehensive Multi-modal Video Understanding Benchmark.

[4] ViLMA: A Zero-Shot Benchmark for Linguistic and Temporal Grounding in Video-Language Models.

[5] Video-MME: The First-Ever Comprehensive Evaluation Benchmark of Multi-modal LLMs in Video Analysis.

[6] Video-ChatGPT: Towards Detailed Video Understanding via Large Vision and Language Models.

**Questions:**

N/A

---

### Official Review · Reviewer_Lsxn · 2024-11-03

**Soundness:** 3
**Presentation:** 2
**Contribution:** 2
**Rating:** 5
**Confidence:** 4

**Summary:**

The paper introduces a new benchmark called TemporalBench, designed to evaluate the fine-grained temporal understanding capabilities of multimodal video models. The benchmark consists of approximately 10,000 pairs of video description questions derived from around 2,000 high-quality human-annotated video captions. TemporalBench provides fine-grained temporal annotations to assess models' ability to reason about time-sensitive events in videos. Besides, the paper shows a pitfall in multi-choice question answering due to the potential for models to exploit answer choice cues and proposes multiple binary QA as a more robust evaluation metric. The results show that even state-of-the-art models like GPT-4o achieve only 38.0% multiple binary QA accuracy on TemporalBench, indicating a significant gap between human and AI capabilities in temporal understanding. The paper highlights the limitations of current video understanding benchmarks and the need for more detailed temporal annotations to improve models' reasoning capabilities. The authors hope that TemporalBench will foster research into improving models' temporal reasoning abilities.

**Strengths:**

The paper proposes TemporalBench to address fine-grained temporal issues, providing constructive results to for subsequent research. Its strength and contributions can be concluded as follows:

1. The benchmark provides detailed temporal annotations that allow for a more nuanced evaluation of models' capabilities, moving beyond coarse-grained annotations found in popular benchmarks like MSRVTT.
2. TemporalBench sources its data from a variety of video datasets, ensuring a wide range of scenarios and actions are covered.
3. The paper shows a pitfall in multi-choice question answering due to the potential for models to exploit answer choice cues and proposes multiple binary QA as a more robust evaluation metric.
4. The paper presents comprehensive experimental results, evaluating the capabilities of multiple multimodal models across various tasks such as video captioning, binary QA, and long video understanding, providing a multidimensional assessment of their performance.

**Weaknesses:**

The paper raises valuable fine-grained temporal issues and proposes to address them with TemporalBench, providing some constructive results for subsequent research. However, there are several points where the paper could be improved:

1. Lack of detailed results on centralized and decentralized captions. The paper presents the percentage of GPT-4o predictions aligned with N1(C) and C in Table 4. I suggest that the paper can provide results of different multimodal models in Table 4. More detailed analysis can make the caption decentralization method more convincing by demonstrating the result across a broader range of models, not just GPT-4o.
2. I suggest that the paper can provide some detailed information (such as the prompts and detailed procedure) on how LLMs are used to generate the benchmark from 2K human-annotated captions.

**Questions:**

Based on the words per second and clip length distribution provided in the paper, it can be inferred that the average caption length for each video clip is around 250-300 words. This significantly exceeds the context length used in works like XCLIP with CLIP models, which is generally 77. I wonder that:

1. Is the long text truncated during the testing of the embedding models? If it is truncated, it indicates that the input is not fully fed into the models.
2. Is there any interpolation of the positional encodings (like what has been done in InternVideo2) to ensure all long texts are fed into the model?

Empirically, increasing the truncation length and interpolating positional encodings can help improve the performance of traditional CLIP models on long caption benchmarks.

---

### Official Review · Reviewer_s93Z · 2024-11-03

**Soundness:** 3
**Presentation:** 3
**Contribution:** 3
**Rating:** 5
**Confidence:** 4

**Summary:**

The paper introduces TemporalBench, a new benchmark for evaluating fine-grained temporal understanding in multimodal video models. It addresses the limitation of current video benchmarks like MSRVTT and TGIF, which fail to effectively evaluate AI models' temporal reasoning due to the lack of detailed temporal annotations. TemporalBench consists of approximately 10K pairs of video description questions derived from around 2K high-quality human-annotated video captions. It provides fine-grained temporal annotations to assess models' ability to reason about time-sensitive events. The benchmark tests models on tasks like video QA, captioning, and long video understanding, and shows that even state-of-the-art models like GPT-4o perform poorly, highlighting a significant gap between human and AI capabilities in temporal understanding.

**Strengths:**

1. TemporalBench improves the evaluation of multimodal video models' spatio-temporal comprehension by providing approximately 10K pairs of video description questions with fine-grained temporal annotations that capture subtleties in video actions and events.

2. The benchmark supports various tasks such as video QA and video captioning and sources data from a range of video domains, ensuring a broad and diverse set of testing scenarios.

3. By benchmarking against human performance, TemporalBench exposes the significant gaps in current state-of-the-art models' fine-grained temporal understanding, indicating areas for future research and development.

**Weaknesses:**

1. One major concern is about the video data sources used in this benchmark. From what I can see, the authors only used existing academic datasets such as COIN, Activity, and EgoExo4D to source their videos, rather than collecting any new video data from more recent real-world scenarios. This is quite problematic because these academic datasets are already widely known and used in the research community. What this means is that many models may have already been trained or fine-tuned on these videos in some way. As a result, model developers could potentially find ways to optimize their models specifically for these familiar videos without too much effort, leading to artificially inflated performance scores that don't actually reflect how well the models would work on truly novel, real-world video content.

2. I found it quite frustrating that the paper doesn't include any comprehensive statistical comparison tables between TemporalBench and other existing video benchmarks. Without such detailed comparisons showing metrics like dataset size, video duration distributions, annotation density, and task complexity, it's really hard for readers like me to get a clear picture of exactly what makes TemporalBench special or better than previous benchmarks. Some simple comparison tables would have made it much easier to understand TemporalBench's unique contributions and advantages.

3. Regarding the MBA evaluation method proposed in the paper - while I appreciate that it creates a more challenging and thorough evaluation scenario, I have some practical concerns. The method requires significantly more computational resources and evaluation time compared to traditional approaches. This raises an important question that the authors haven't fully addressed: how do we find the right balance between having a rigorous evaluation process and keeping the computational costs manageable? This is especially relevant for researchers with limited computing resources.

4. A significant limitation of this work is that the dataset hasn't been made publicly available yet. This is quite disappointing because it prevents other researchers from independently verifying the results or building upon this work. The research community would benefit tremendously if the authors could release the complete dataset, including all the videos, annotations, and evaluation scripts. This would allow for more transparent validation of the benchmark's claims and accelerate progress in temporal understanding research. I strongly encourage the authors to consider making their dataset open-source as soon as possible.

**Questions:**

1. When will the dataset be open-sourced?

2. How to balance the evaluation cost and evaluation effect?

---

### Official Review · Reviewer_zsqf · 2024-11-04

**Soundness:** 3
**Presentation:** 2
**Contribution:** 2
**Rating:** 3
**Confidence:** 4

**Summary:**

In this paper, the authors introduce a new benchmark for evaluating the fine-grained temporal understanding capabilities of multimodal video models. The authors argue that existing video benchmarks lack the necessary fine-grained temporal annotations to effectively evaluate MLLMs' temporal reasoning abilities. To address this, they present TemporalBench, which  provides fine-grained temporal annotations to assess models' abilities to reason about temporal sequences. The results show that even state-of-the-art models like GPT-4o perform poorly on TemporalBench, highlighting a significant gap in temporal understanding between humans and MLLMs.

**Strengths:**

- The paper is clearly written, making it easy to follow the authors' arguments and results.
- The benchmark is of high quality, featuring a diverse set of video clips and detailed human annotations refined by the authors.

**Weaknesses:**

- The figures and tables are not well-organized. For example, Figure 2 should be on page 4, and Table 1 should be on page 5.
- Although the authors claim that the benchmark can be used for "Dense video-text matching and retrieval", "Video grounding from detailed text descriptions", and "Text-to-Video (T2V) generation with detailed prompts", there are no related results presented.
- The details of generating negative caption candidates are omitted, which makes reproduction difficult. More importantly, this replacement strategy is similar to VideoCon [1], yet the authors do not cite it.
- The paper lacks in-depth analysis. Although the authors show that current MLLMs do not perform well in TemporalBench, there is no detailed analysis explaining why they are unsatisfactory or how they can be improved.
- The authors use outdated metrics for video captions, such as BLEU, which are not very suitable for MLLMs due to their long responses.

---

Reference:

[1] Bansal H, Bitton Y, Szpektor I, et al. Videocon: Robust video-language alignment via contrast captions[C]//Proceedings of the IEEE/CVF Conference on Computer Vision and Pattern Recognition. 2024: 13927-13937.

**Questions:**

See weaknesses

---

### Official Review · Reviewer_Eio5 · 2024-11-05

**Soundness:** 3
**Presentation:** 2
**Contribution:** 2
**Rating:** 5
**Confidence:** 4

**Summary:**

Motivation:  Popular video benchmarks, such as MSRVTT and TGIF, often fail to effectively evaluate AI models’ temporal reasoning abilities due to the lack of fine-grained temporal annotations.
They  introduce a new TemporalBench benchmark for fine-grained temporal event understanding in videos. TemporalBench, sourced from a diverse video datasets, consists of ∼10K pairs of video description questions, derived from ∼2K high-quality human-annotated video captions.
Their results show that state-of-the-art models like GPT-4o achieve only 38.0% multiple binary QA accuracy on TemporalBench, demonstrating a significant human-AI gap in temporal understanding.

**Strengths:**

1. Their benchmark enriches several fundamental video understanding tasks due to its detailed captions.
2. The experimental results showcase the specific performance of various video understanding models, accompanied by corresponding result analysis.
3. There are some interesting findings regarding Multiple Binary QA.

**Weaknesses:**

1. There is a lack of detailed comparison with the temporal understanding aspect in the following paper: MVBench: A Comprehensive Multi-modal Video Understanding Benchmark. Additionally, it lacks a specific definition for temporal understanding tasks compared to this benchmark.
2. It is unclear what specific example questions are used for the "binary QA accuracy (BA)" and "multiple binary QA settings" in Table 2.
3. This seems more like a hallucination-reduction captioning benchmark rather than a temporal benchmark.
4. The approach description and analysis for generating negative captions are insufficient. The types of negatives will determine how different models perform on this benchmark, and the benchmark results also lack corresponding analysis.

**Questions:**

See above.

---

### Note · Authors · 2024-11-14

I have read and agree with the venue's withdrawal policy on behalf of myself and my co-authors.